# Effects of Using a Shoulder/Scapular Brace on the Posture and Muscle Activity of Healthy University Students during Prolonged Typing—A Randomized Controlled Cross-Over Trial

**DOI:** 10.3390/healthcare11111555

**Published:** 2023-05-25

**Authors:** Melissa Leung, Mandy M. P. Kan, Hugo M. H. Cheng, Diana E. De Carvalho, Shahnawaz Anwer, Heng Li, Arnold Y. L. Wong

**Affiliations:** 1Department of Rehabilitation Sciences, The Hong Kong Polytechnic University, Hong Kong SAR, China; melissa.leung@connect.polyu.hk (M.L.); mandy.kan@polyu.edu.hk (M.M.P.K.); 16118487g@connect.polyu.hk (H.M.H.C.); arnold.wong@polyu.edu.hk (A.Y.L.W.); 2Faculty of Medicine, Memorial University of Newfoundland, St. John’s, NL A1B 2Y1, Canada; diana.decarvalho@med.mun.ca; 3Department of Building and Real Estate, The Hong Kong Polytechnic University, Hong Kong SAR, China; heng.li@polyu.edu.hk

**Keywords:** typing, scapular brace, ergonomics, neck and shoulder pain, fatigue

## Abstract

Laptop use appears to contribute to poor working postures and neck pain among university students. Postural braces have the potential to improve upper back/neck posture and therefore might have a role as an ergonomic aid for this population. Therefore, the purpose of this study was to assess the short-term effects of scapular bracing on pain, fatigue, cervicothoracic posture, and the activity of the neck and upper-back muscles in healthy college students. A randomized controlled crossover trial was conducted to evaluate the self-reported pain and fatigue, the amplitude and median frequency of surface electromyography in neck extensors, upper trapezius, and lower trapezius, as well as the neck and shoulder sagittal alignment (measured by inertial sensors and digital photographs) during a 30-min typing task in a sample of young, healthy university students with or without a scapular brace. The brace condition resulted in significantly smaller levels of bilateral trapezius muscle activity (*p* < 0.01). Rounded shoulder posture was slightly better in the brace condition, but these differences were not significant (*p* > 0.05). There were no significant immediate differences in pain or fatigue scores, neck alignment, or the electromyographic activity of the other muscles tested between brace and non-brace conditions (all *p* > 0.05). However, bracing appears to immediately reduce the electromyographic activity of the lower trapezius muscles (*p* < 0.05). These findings shed some light on the possible advantages of scapular bracing for enhancing laptop ergonomics in this group of individuals. Future studies are warranted to evaluate the effects of different types of braces, the importance of matching the brace to the user, and the short- and long-term effects of brace use on computer posture and muscle activity.

## 1. Introduction

With the increased use of laptop computers and computer-based learning, university students spend significant periods of time seated [1,2]. Prolonged deskwork in a poor sitting posture is thought to heighten the risk of developing musculoskeletal problems in this population [2,3,4]. Cross-sectional research has reported that 13% to 46% of undergraduate students report having new neck pain within a one-year period, with 33% of them experiencing persistent pain [5]. Studies have also revealed a positive association between the duration of computer use and self-reported musculoskeletal discomfort among university students [6,7,8]. Given the large global burden of musculoskeletal disorders and their associations with health-related quality of life and depression [9,10], the presence of such pain may hinder the academic and extracurricular performance of students.

It has been suggested that a component of neck pain is attributed to exposures to non-neutral spine alignment, eliciting excessive neck and shoulder muscle activation and increased cervical loading or upper trunk flexion [11,12]. Posture-related alterations in motor unit recruitment patterns may also increase fatigability and histochemical changes in muscles, resulting in pain [13]. Slumped sitting can increase the strain on the spine [1], and is characterized by increased thoraco-lumbo-pelvic angulation [14], forward head posture (FHP), rounded shoulder position (RSP), and increased cervical erector spinae (CES) activities [11,15]. Choi and colleagues [16] showed that individuals sitting in a more hunched position presented with significantly higher levels of muscle fatigue in the splenius capitis and upper trapezius (UT). Theoretically, maintaining a neutral spine position helps evenly distribute biomechanical load across surrounding structures and reduces postural strain in the neck [1]. Therefore, effective management strategies are important to minimize posture-related strain on neck or shoulder muscles.

The development of an effective and affordable neck/shoulder pain intervention could enhance the quality of life and work productivity of these patients, as well as lower their medical costs [17]. Lee and colleagues [18] found that scapular posterior tilting exercises alongside shoulder bracing were significantly better than exercises alone in reducing RSP among young male adults. To date, only one study has investigated the effect of using a shoulder/scapular brace on altering posture and muscle activity in university students [19]. These authors found that brace application significantly reduced forward shoulder angle (FSA) and increased lower trapezius (LT) muscle activity. However, because the study recruited diverse overhead sports athletes and collected data during the performance of various upper limb exercises, the results cannot be generalized to other university students, who may also be at risk of developing neck and shoulder pain due to poor computer posture. Postural correction using thoracic and scapular taping could significantly increase shoulder flexion and abduction in people with shoulder impingement syndrome [20]. Recently, Chiu et al. [21] showed that the characteristics of a shoulder brace strap could affect muscle activity and scapular kinematics at various arm angles. The clinical preference for treating shoulder impingement syndrome and rounded shoulder pain recommends using a shoulder brace with a diagonal construction that imposes tension. The use of a shoulder brace with maximum strap tension has been shown to improve rounded shoulders [22,23]. Previous research has also revealed that wearing compressive sleeves and clothing consistently improves both active and passive joint repositioning sensations [24,25]. They hypothesized that the cutaneous feedback from compression clothing would significantly improve proprioception. Similarly, Uhl et al. [26,27] found that the use of one type of commercially available scapular brace led to an increase in posterior tipping, a decrease in upward rotation in both the dominant and nondominant upper extremities, and a decrease in internal rotation during the lowering phase of arm elevation. Although many available shoulder/scapular braces on the market claim that they can correct posture, it remains uncertain whether they can improve users’ posture and muscle activity during computer use.

To address these gaps in the literature, the current study aimed to quantify the immediate effects of scapular bracing on pain, fatigue, cervicothoracic posture, and the muscle activity of the neck and upper back among university students during a 30-min exposure to a computer typing task. It was hypothesized that the use of scapular bracing would cause less pain and fatigue, less FHP and RSP, and less CES and UT activity but higher LT activity compared to individuals without scapular bracing. The results would provide preliminary evidence to support the use of this approach to improve the computer posture of university students.

## 2. Methods

### 2.1. Participants

Healthy university students aged 18 years or older were recruited by convenience sampling through posters on a university campus. The exclusion criteria were prior spine surgery, previously diagnosed spine or shoulder pathology, injuries in neck and shoulder areas within six months before the study, undergoing spine or shoulder rehabilitation, previous experience with using a shoulder or scapular brace, or Neck Disability Index scores above 14. All participants provided written informed consent. The study was approved by the Human Subjects Ethics Sub-committee of the Hong Kong Polytechnic University (HSEARS20180624001).

By assuming a medium effect of the brace on the FHP and upper trapezius muscle activity (η^2^ = 0.30) [19], the calculated sample size using G*Power 3.1.9.7 (Heinrich-Heine-Universität Düsseldorf, Düsseldorf, Germany) was 30 for a crossover study design with a statistical power of 0.80 and an alpha level of 0.05 (two-tailed).

### 2.2. Experimental Design and Procedure

This was a crossover randomized controlled trial study design, which involved a single laboratory visit. After providing consent, participants completed a set of self-reported questionnaires.

The participant then sat on a wooden stool without armrests, with their knees and hips flexed at approximately 90° and their elbows at 70° to 80° (Figure 1). The wooden stool was chosen because many local students sit on benches without backrests to use laptop computers. Further, many students do not lean against the backrests of chairs when they use laptop computers. Each participant was instructed to type on a free online typing test website (https://keyhero.com/free-typing-test/) (accessed on 1 April 2020) for one minute on a laptop computer to familiarize himself/herself with the task and the attached sensors. The participant was then randomized (in a block of six) into one of two sequences (bracing and then non-bracing, or non-bracing and then bracing) to perform a 30-min typing task. Specifically, research personnel unrelated to this study used an Excel file to generate random numbers and put them into sequenced opaque envelopes. A researcher blinded to the randomization process opened the envelope to allocate participants.

The 14″ laptop computer was placed 3 cm from the edge of the desk, while the screen-to-keyboard angle was 120°. The participants were allowed to rest their forearms and hands on the desk and laptop computer during the typing task. The typing speed (words per minute) was measured by the typing program. At each 10-min interval of the data collection, participants were reminded of the remaining time. A 30-min typing task was chosen because prior research found that sedentary time in bouts of less than 30 min was recommended because it was associated with a lower risk of all-cause mortality as compared to longer sedentary time [28]. This duration has also been used in other sitting-related ergonomics studies [29,30]. Further, two shoulder-taping studies found significant decreases in the UT/LT activation ratio during shorter durations of typing [31,32]. After the first typing task, a 30-min washout break was given, during which the participant could choose to sit or lie down. The participants repeated the second 30-min typing task with the other bracing condition.

For the trial with bracing, the participant put on a medium-sized scapular brace, BACK Posture Hero™ (Handsome Company Ltd., Manchester, UK), according to the manufacturer’s recommendations and fitted it to the point of subjective tightness without discomfort. The brace comprises a flexible plastic spine to extend the thoracic spine and two Velcro straps to bring the shoulders backward. The brace was put on like a backpack, with the plastic spine pressed against the thoracic spine. The two extensible straps ran from the superior plastic spine in a posteroanterior direction over the upper trapezius through the axilla posteroinferiorly to the inferior plastic spine, where the straps changed direction to wrap around the rib cage anteriorly and were fixed at the sternal region by the Velcro on the straps. The brace extended the thoracic spine and retracted the bilateral shoulders (Figure 2). Participants were given 15 min to adapt to the new sensation prior to the start of the familiarization trial.

### 2.3. Measurements

#### 2.3.1. Subjective Pain and Fatigue

Participants at pre- and post-trial were asked to report their subjective pain scores and rate of perceived exertion (RPE) using the Numeric Pain Rating Scale (NPRS) and modified Borg RPE Scale, respectively [33,34]. To minimize the influence of discomfort contributed by factors other than the brace (e.g., discomfort from the wooden stool), participants were instructed to focus on their neck and shoulder symptoms only.

The NPRS is an 11-point ordinal scale, ranging from 0 (no pain at all) to 10 (the worst imaginable pain). A change of two points is considered to be the minimal clinically important difference (MCID) [35]. Similarly, the modified Borg RPE scale is an 11-point ordinal scale ranging from 0 (no exertion at all) to 10 (maximal exertion). The MCID is a one-point change [36].

#### 2.3.2. Surface Electromyography (sEMG)

Six pairs of surface electromyography (sEMG) electrodes were then attached to bilateral cervical and back muscles (CES, UT, and LT) to evaluate whether the brace could alter neck and shoulder muscle activity during the typing task. The sEMG activities of these muscles during two maximum voluntary isometric contractions (MVIC) in three specific positions were performed. The highest MVIC sEMG activities were then used for subsequent sEMG normalization (Table 1). Following the MVIC, four wearable inertial measurement units (IMUs) (Noraxon MyoMotion, Noraxon USA Inc., Scottsdale, AZ, USA) were affixed midline at the back of the head, C7, T12, and S2 levels. During the typing trial, the participant’s neck and back muscle activities were measured by sEMG.

##### Amplitude

To measure the muscle activities of bilateral CES, UT, and LT, the skin overlying the target muscles was briskly wiped with an alcohol swab and sandpaper to ensure that skin impedance was at ≤5000 Ω. Disposable bipolar Ag/AgCl surface electrodes with a 10 mm active diameter (3M, Minnesota, MN, USA) were attached along the direction of the target muscle fibers (Table 1) with an inter-electrode distance of 20 mm [37,38].

The sEMG signals were recorded at 1500 Hz by a wireless sEMG system (Telemyo, Noraxon Inc., Phoenix, AZ, USA; CMRR 100 dB at 60 Hz) and digitized with a desktop direct transmission system (Noraxon Inc., Phoenix, AZ, USA). Raw signals were processed by Noraxon MR3.10.2 software to eliminate electrocardiographic signals. A notch filter at 50 Hz and a bandpass filter between 10 and 500 Hz were applied to remove electrical noise and estimate target muscle activity, respectively [39]. The root mean square (RMS) sEMG signals of each muscle over a 100-millisecond moving window throughout the two 30-min conditions were calculated, normalized to the corresponding RMS sEMG during MVIC, and expressed as %MVIC. The 50th percentile of an amplitude probability distribution function (APDF) was used to determine the median sEMG signals of each muscle during the bracing or non-bracing condition, as suggested in prior research [40].

Two 5 s MVIC tests of each target muscle against manual resistance were performed according to the established protocols (Table 1) [33], while the respective sEMG signals were recorded. One familiarization practice with submaximal manual resistance was performed for each testing position. A 1-min rest was given between MVIC tests to minimize fatigue [41]. The maximum RMS sEMG signal of a given muscle was identified using a 100-millisecond moving window passing through each MVIC trial. This value was used to normalize sEMG signals during the typing tasks.

##### Median Frequency (MF)

The MFs of the sEMG power spectrum of each target muscle in the 1st, 15th, and 29th minutes were analyzed by the Fast Fourier Transform technique with a smoothing Hamming window digital filter [42,43]. The MFs of sEMG at the 1st, 15th, and 29th minutes were then normalized to the MF obtained in the first minute. These normalized MFs were plotted against time to estimate the slope. A negative slope on the normalized MF plot of a given muscle indicated myoelectric muscle fatigue.

#### 2.3.3. IMUs

The relative changes in spine angles (spinal posture) were captured by IMUs. The IMUs were calibrated with the participant seated in an upright position at the beginning of each trial. The kinematic data were sampled at 50 Hz during the trials. The kinematic data were processed using the Noraxon MR3.10 program to determine the degrees of angular change occurring between two adjacent sensors, which indicated a spinal region (Table 2) [44].

#### 2.3.4. Digital Photographs

The temporal changes in neck and shoulder postures (i.e., forward head angles (FHAs) and FSAs) were captured by a high-resolution camera. The FHA and FSA during the typing task were measured using a method suggested by Perry et al. [45]. Specifically, a camera (Cybershot DSCH50; Sony, Tokyo, Japan) was mounted upon a tripod at a height of 80 cm and placed at a horizontal distance of 250 cm left of the seat. A red sticker was attached to the participants’ left acromion as a point of reference for the estimation of FSA. FHA was defined as the angle between the vertical plumb line through C7 and the line connecting C7 and the tragus. FSA was defined as the angle between the vertical plumb line through C7 and the line connecting C7 to the center of the acromion (Figure 3) [46]. Photographs of the participant’s sagittal head, neck, and shoulder postures were captured at the 1st, 15th, and 30th minutes of each trial. Photos were processed using Adobe Photoshop CC (Adobe, San Jose, CA, USA) to measure FHA and FSA [45,46]. A researcher unrelated to the study randomly numbered the photos of each participant before measurement to minimize bias. A trained physiotherapy student conducted the measurements.

### 2.4. Statistical Analysis

All statistical analyses were conducted using SPSS (version 25.0; SPSS Inc., Chicago, IL, USA). The significance level was set at 0.05. The normality of the data was determined by Shapiro—Wilk tests. Separate generalized estimating equations (GEE) were used to compare differences in pain and fatigue scores, MF slopes of sEMG signals, and spinal angles between the two conditions because these data were non-parametric. A Wilcoxon signed-rank test with Bonferroni correction was used for post-hoc tests, if applicable. Further, the Wilcoxon signed-rank test was used to compare the 50% APDF of sEMG between the two conditions. FHA and FSA were compared using 2-way repeated measures analyses of variance because these data were normally distributed. The within-subject factor was the timepoints. Effect sizes were calculated as partial eta squared (η^2^p) and then converted to Cohen’s *d*, which is described as small (0.2), medium (0.5), or large (0.8) [47].

## 3. Results

Fifteen males and fifteen females (mean age: 25.6 ± 1.7 years; weight: 59.0 ± 8.8 kg; height: 21.7 ± 1.8 m) were recruited. Their mean typing speed was 51.4 ± 17.7 words per minute (wpm), ranging from 30.7 to 108 wpm. No participants dropped out.

### 3.1. Subjective Fatigue/Pain Scores

There were significant increases in the mean NPRS (mean = 0.8 ± 0.2) and RPE scores (mean = 1.8 ± 0.3) immediately after completing the 30-min typing task with the brace (*p* < 0.05). Similarly, there were significant increases in the mean NPRS (mean = 0.8 ± 0.2) and RPE scores (mean = 2.4 ± 0.3) at post-trial without a brace (*p* < 0.05). However, there was no significant between-condition difference (Table 3).

### 3.2. 50% APDF

All muscles showed slightly greater absolute values of 50% APDF of %MVIC during the non-bracing condition, except for the right UT (Table 4). Only bilateral LT muscles under the non-brace condition had a significantly higher 50% APDF of %MVIC than the bracing condition (*p* = 0.008). The mean 50% APDF of %MVIC ranged from 1.68% to 12.17% during typing with a brace and 1.71% to 12.18% without a brace.

### 3.3. MF Slopes

There was no significant between-condition difference in the MF slope of EMG signals across the six muscles (Table 5). Only the right UT displayed negative slopes under both conditions (slope: −0.15%, −0.09%) and the left CES in the brace trial (slope: −0.05%), indicating muscle fatigue.

### 3.4. IMU Measurements

There was a significant reduction in cervical spine flexion angles between the first and fifteenth minutes under the non-bracing condition. Compared to the bracing condition, the non-bracing condition had smaller cervical flexion angles across all timepoints (Table 6), but the largest between-condition difference was only 2°.

### 3.5. FHA and FSA

At baseline, the average FHA and FSA under the bracing condition were 51.0° and 45.6°, respectively. Likewise, the average FHA and FSA under the non-bracing condition were 50.6° and 52.0°, respectively. Across the three timepoints, wearing a brace demonstrated a larger absolute FHA and a smaller absolute FSA than the non-bracing condition (Table 6). This means that while their shoulders were less rounded, participants with the brace needed to bend their neck more, which could introduce more focal forces at the cervicothoracic junction. However, the differences between the two conditions were not statistically significant.

## 4. Discussion

Bracing or taping the scapulothoracic articulation is one way to restore normal scapular posture and scapular muscle activation. Scapular taping has been shown to change the scapular posture, reduce UT muscle activity, and improve pain profiles in patients with shoulder impingement syndrome [20,48,49,50]. However, adhesive tape may have the risk of irritating the skin of some individuals, making it unsuitable for long-term or daily use [19]. Companies have created braces based on this concept to help patients with shoulder issues achieve better scapular posture and muscular activity. These braces aim to improve scapular position, muscular activation, and motion by modifying the shoulder and thoracic spine posture [19]. Walther et al. [51] compared the efficacy of a functional brace to that of conventional rehabilitation and home-based programs for treating subacromial impingement syndrome. After 6 and 12 weeks of treatment, the braced group experienced the same reductions in shoulder discomfort and gains in function as the guided self-training program or the traditional physiotherapy group. They concluded that splinting had the potential to be as beneficial as other common treatments for shoulder impingement syndrome [51]. Therefore, bracing may be a novel method for repositioning the scapula and relieving shoulder pain in patients with various disorders. Scapular braces are widely used by athletes undergoing rehabilitation for shoulder problems [19]. In the clinical setting, athletic trainers may use bracing or taping as an adjunct intervention to a corrective exercise program [48,49]. Alternatively, athletic trainers may use bracing or taping during corrective exercise programs to reestablish more normal length–tension relationships in the shoulder muscles [50]. As such, there is a pressing need for evaluating the effects of shoulder braces on the postural correction of healthy computer users who are at risk of developing rounded shoulders. The current randomized controlled cross-over trial was conducted to determine the effect of a commercially available scapular brace on pain, fatigue, and biomechanics of the neck and shoulders during a typing task that mimicked the intensity and duration of computer use by typical university students. Although previous research has examined the effects of a scapular brace on the posture and activation of scapular muscles during resisted overhead tasks [19], these movements are uncommon among university students.

### 4.1. Subjective Neck Pain/Fatigue

There were no statistically significant differences between conditions in NPRS and RPE scores, although the non-bracing condition had larger absolute temporal increases in both scores. Interestingly, there was a clinically significant increase in the mean modified Borg RPE score (2.43 points) under the non-bracing condition, which was larger than the MCID for the RPE scale [52]. The post-typing increase in NPRS score was less than one point in both conditions, which was smaller than the MCID for NPRS [13]. It is noteworthy that during the experiment, some individuals verbally expressed discomfort associated with wearing the brace. Commonly reported areas of discomfort were bilateral UT regions, where the posterior straps cross from the back to the anterior shoulders. Another discomfort site was the mid-thoracic region, as the brace’s back support contained ridges that contacted the participant’s mid-thorax to facilitate extension. This observation suggests that novel cutaneous stimulation from the brace might cause some discomfort to wearers.

### 4.2. sEMG Activity and APDF

Prior studies have shown that higher typing speeds are associated with greater muscle activity, and a task that requires participants to maintain a fast typing speed may induce more muscle fatigue [31,53]. Because our participants were instructed to type at their usual speeds, our trial might not be intense enough to elicit significant discomfort that might benefit from the use of a brace. The low physical demand of our task was substantiated by our relatively low 50% APDF of sEMG values as compared to those reported from research involving more demanding computer tasks [54]. Further, significant negative MP slopes were only noted in the right UT and left CES, indicating that only these two muscles showed signs of fatigue [55].

The significantly higher bilateral LT activity during the non-bracing trial than the bracing trial was unexpected. Previous research has found that symptomatic individuals with chronic shoulder impingement tend to have a higher UT/LT activation ratio on the affected side [56]. As UT tends to be over-activated during typical typing, an ideal ergonomic intervention is to reduce UT activity and increase LT recruitment [31]. Maintaining a UT/LT activation ratio close to one may help lower the risk of overcompensation by UT, which may elicit abnormal scapulohumeral rhythm [57]. Our results revealed that bracing did not alter the UT/LT activation ratio in asymptomatic individuals. Although speculative, the significant reduction in bilateral LT activity during the brace condition might be attributed to the application of a compressive downward force of the brace straps on bilateral levator scapulae. As one function of LT is to stabilize the scapula by counterbalancing the scapular elevation force generated by levator scapulae [58], the presence of downward compression on levator scapulae by the straps will help stabilize the scapula and lead to reduced LT activity. Future studies should clarify the potential beneficial effects of scapular braces on shoulder muscle fatigue following a prolonged typing task.

The lack of systematic between-condition differences in 50% APDF of sEMG and MF slope might be attributed to the muscle redundancy phenomenon. This phenomenon occurs because there are infinite combinations of muscle activation patterns that can be orchestrated by the brain to accomplish a given task with a minimal risk of muscle fatigue [25]. Although humans generally adopt stereotypical activation patterns during typing, there are great individual variations in muscle activation patterns. Everyone rarely replicates the exact same pattern twice. Therefore, great within- and between-individual variabilities in muscle activation patterns may lead to the observed non-significant results.

### 4.3. Motion Analysis, FHA, and FSA

Under the non-bracing condition, the mean cervical flexion angle of participants as measured by IMUs significantly decreased by 2.3° at 15 min. This magnitude was larger than the reported IMU error (0.3°) [59]. Conversely, there were no significant temporal changes in cervical flexion angles in the brace condition across all timepoints, suggesting that wearing a scapular brace for 30 min may not be effective in reducing FHA. This finding concurred with Cole and colleagues’ results [19], which found no significant reduction in FHA when performing overhead arm movements with a scapular brace. Although the tasks and participants’ characteristics differed between the two studies, both studies consistently showed no significant alterations in FHA. This might be because the straps of the brace in both studies were mainly wrapped around the mid-torso and scapular regions. Therefore, these braces might not have any direct corrective forces on the cervical region. That said, the FHA can also be affected by the relative curvature of the thoracic and/or the lumbar spine in sitting [60], which can be influenced by the human machine interface (e.g., the relative position between the participant and the computer, or between the participant and the furniture), types of furniture (e.g., with or without back support, type of chairs, or table height), and the characteristics of the computer (e.g., screen height, screen size, or types of keyboards). Therefore, our findings should be interpreted with caution.

According to Thigpen et al. [50], FHA ≥ 36° and FSA ≥ 52° are the thresholds for FHP and RSP, respectively. Although the differences in FHA and FSA between the two conditions were not significant, it was noteworthy that the mean FSA under the non-bracing condition exceeded 52° at the first two timepoints, while that under the bracing condition did not exceed the threshold at any timepoints. This implies that the brace may have the potential to correct the shoulder alignment of people with mild RSP. Given that the use of compressive neoprene shoulder bracing could significantly improve shoulder joint repositioning sense [61], our brace might also generate similar proprioceptive inputs that helped temporarily correct our participants’ FSA. This could imply that braces may have beneficial effects on posture through both mechanical re-alignment of the joints and increased proprioception [19]. Our findings lend credence to the idea that bracing can temporarily alter shoulder posture. This improvement appears to be due to a combination of mechanical shifts in shoulder girdle position and increased proprioceptive feedback from the brace [19]. Researchers in the fields of medicine and sports science have shown that the use of compressive sleeves and clothing like the brace used in this study consistently enhances both active and passive joint repositioning sense [24,25]. Compression garments or braces are being advocated for their ability to significantly improve proprioception by way of cutaneous feedback [24]. Compression garments, sleeves, braces, or tape applied to the shoulder and scapular area can change proprioception and static posture [25]. However, given the small sample size and no evaluation of neck/shoulder proprioception, our hypothesized mechanism should be verified in future studies.

### 4.4. Limitations and Future Research Directions

The current study has some limitations. First, because the straps of the brace lie near the location of the attached EMG electrodes of the target muscles, they might have compressed the surrounding skin or muscles and influenced the recorded EMG signals. However, this risk was minimal because the examiner carefully placed the surface electrodes to minimize this error. Second, the tightness of the brace was subjectively determined by each participant; hence, the relative force applied by the brace varied among participants. This might have affected the observed muscle activation patterns. However, because prior brace and taping studies found that muscle activation patterns were comparable regardless of the tension applied to the body [19,32], this confounding effect might be minimal. Third, because our participants were asymptomatic university students, the findings cannot be generalized to younger or older populations or those with clinical conditions. Fourth, our 30-min washout period between trials was selected based on prior ergonomic studies in which the washout period varied from 2.5 min to 40 min [62]. Although we did not control participants’ activities during the resting period, it might have affected the recovery between trials. That said, no participants complained about pain or fatigue before the second typing task. Fifth, there was also the problem that just one type of brace was attempted. Because different brace applications may have different effects on posture and EMG, trying a new approach may be the best option. Sixth, while we made all efforts to blind the participants to safeguard the validity of the results, the outcome evaluators were not blinded, so we cannot rule out the possibility that the participants changed their posture and muscle activity simply because they were taking part in the research. Seventh, the current protocol was not pre-registered on a clinical trials registry. Finally, our participants sat on a wooden stool without a backrest during our data collection. However, many individuals may prefer to sit on a chair with back support when they work at a desk or study in a classroom. Therefore, our results may not be generalized to these situations. Further research should directly compare the postural effects of employing a shoulder/scapular brace during a computer task on chairs with and without a backrest.

### 4.5. Ergonomic Implications

Our findings suggest that our chosen commercial scapular brace may not have significant immediate effects on improving shoulder and neck postures or reducing UT activity among laptop users. Given that this pilot study only evaluated the effects of one type of scapular brace on asymptomatic university students, future research should test braces of different designs on asymptomatic and symptomatic populations to evaluate their impacts on the respective subjective feelings and body biomechanics. Further prospective research should examine if the regular use of bracing for several weeks or months can have beneficial ergonomic effects during typing or reading. Since previous research has suggested that taping and compression garments may benefit neck and shoulder biomechanics during upper limb tasks, future studies should compare the effects of different ergonomic interventions on subjective and objective improvements during office tasks [19,32,63]. Given the popularity of tablets and smartphones among university students, it is paramount to investigate the effects of scapular braces on students’ biomechanics during the use of these electronic devices. Additionally, future clinical trials should be conducted to determine the efficacy of brace use as a treatment alternative in addition to regular physical exercise with an emphasis on body posture.

## 5. Conclusions

This study aimed to explore the short-term effects of scapular bracing on the computer posture and muscle activity of young, healthy university students while typing. Bilateral lower trapezius muscle activity significantly reduced under the brace condition. Round shoulder posture was slightly improved under the brace condition, although these differences were not statistically significant. However, comparing the brace and non-brace conditions, there were no significant differences in pain or fatigue scores, neck alignment, or the activity of the other neck or shoulder muscles. The results suggest some potential benefits of scapular bracing for improving the ergonomics of laptop use in this population. However, further research is warranted to investigate the effects of different types of braces, the importance of matching braces to users, and the short- and long-term effects of brace use on computer posture and muscle activity.

## Figures and Tables

**Figure 1 healthcare-11-01555-f001:**
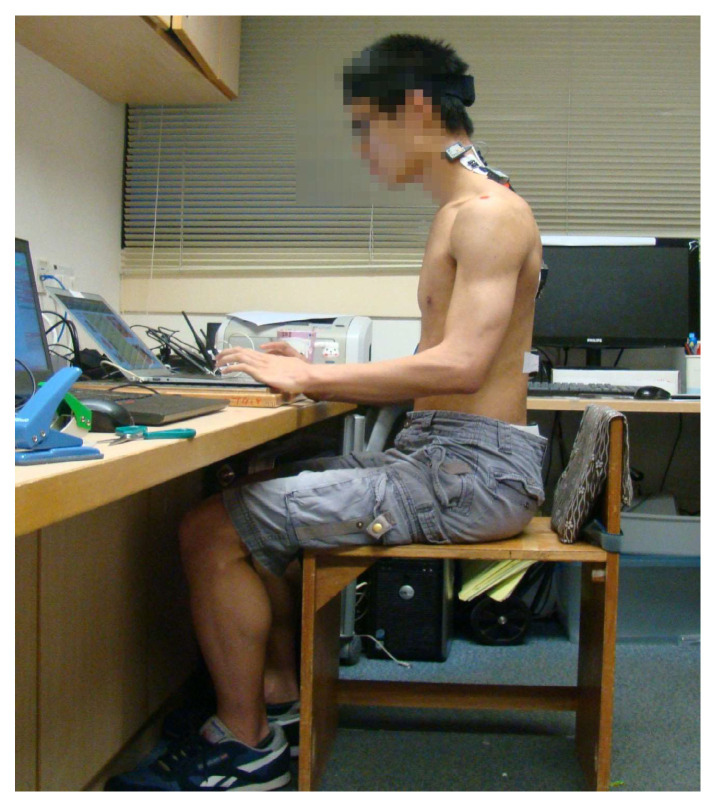
Posture at baseline; participant’s knees flexed at approximately 90° and elbows flexed to 70–80°.

**Figure 2 healthcare-11-01555-f002:**
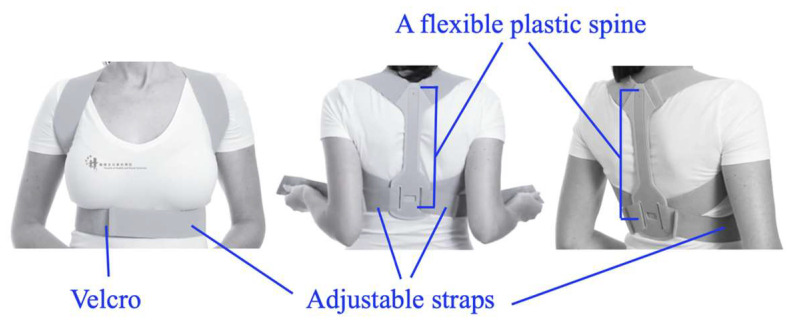
A commercially available scapular brace.

**Figure 3 healthcare-11-01555-f003:**
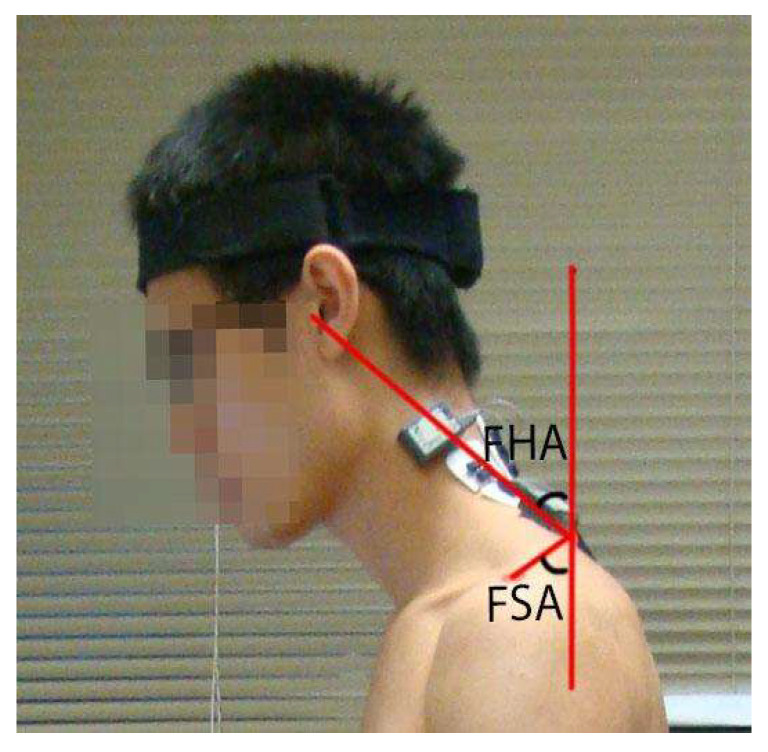
Calculation of Forward Head Angle (FHA) and Forward Shoulder Angle (FSA). FHA was determined by measuring the angle formed by a plumb line drawn vertically through C7 and a diagonal line originating from the tragus of the ear. FSA was determined by measuring the angle formed between the plumb line and a diagonal line originating from the left acromion process.

**Table 1 healthcare-11-01555-t001:** The placement locations of bipolar surface electromyography electrodes on target muscles (i.e., bilateral cervical erector spinae, upper trapezius, and lower trapezius); corresponding procedures for testing maximal voluntary isometric contraction of each muscle.

Testing Order	EMG Electrode Landmarks	MVIC Positioning
1. Cervical Erector Spinae (CES)	2 cm lateral from the C4 spinous process	A participant sat with arms by the side of the body and performed maximal neck extension against manual resistance at the back of the head, applied by an examiner.
2. Upper Trapezius (UT)	Midway between the mastoid process and root of the spine of scapula	A participant sat with arms beside the body, and then performed maximal shrugging of shoulders against an examiners’ downward resistance applied to the acromion.
3. Lower Trapezius (LT)	2 finger-widths medial to the inferior angle of the scapula, at a 45° angle towards the T10 spinous process	A participant in prone lying with arms lifted overhead, in line with lower trapezius muscle fiber orientation. The participant raised the arms towards the ceiling using maximal effort against the examiner’s downward force applied just distal to the humeroulnar joint line.

**Table 2 healthcare-11-01555-t002:** Inertial measurement units and corresponding spinal regions.

Motion Sensor Location	Regional Angle
Back of Head + C7	Cervical Spine
C7 + T12	Thoracic Spine
T12 + S2	Lumbar Spine

**Table 3 healthcare-11-01555-t003:** Means of numeric pain rating scale (NPRS) and Borg rate of perceived exertion (RPE) scores.

	Brace		No Brace		Between-Condition Cohen’s *d*
	Mean ± SE	95% CI	Mean ± SE	95% CI	
*Mean NPRS scores*					
Pre-trial	0.30 ± 0.15 *	0.003, 0.60	0.23 ± 0.11 *	0.0004, 0.47	0.06
Post-trial	1.07 ± 0.29 *	0.48, 1.65	1.03 ± 0.26 *	0.51, 1.56	
*Mean RPE scores*					
Pre-trial	0.67 ± 0.20 *	0.26, 1.07	0.47 ± 0.16 *	0.13, 0.80	0.16
Post-trial	2.50 ± 0.40 *	1.69, 3.31	2.90 ± 0.40 *	2.09, 3.71	

* Significant within-group difference (*p* < 0.05); CI = confidence interval; SE = standard error.

**Table 4 healthcare-11-01555-t004:** Percentage of Maximal Voluntary Isometric Contraction (MVIC) at 50% Amplitude Probability Distribution Function (APDF) for the six muscles.

	Brace		No Brace		Between-Condition Cohen’s *d*
	Mean ± SE	95% CI	Mean ± SE	95% CI	
%MVIC at 50% APDF					
Left erector spinae	10.98 ± 1.15	10.57, 11.39	11.52 ± 1.48	10.99, 12.05	0.07
Right erector spinae	12.17 ± 1.82	11.52, 12.82	12.18 ± 2.08	11.44, 12.92	0.02
Left upper trapezius	1.68 ± 0.17	1.62, 1.74	1.71 ± 0.21	1.63, 1.79	0.02
Right upper trapezius	2.19 ± 0.36	2.06, 2.32	2.17 ± 0.33	2.05, 2.29	0.01
Left lower trapezius	3.74 ± 0.54 *	3.55, 3.93	4.25 ± 0.57 *	4.05, 4.45	0.07
Right lower trapezius	4.15 ± 0.49 *	3.97, 4.33	4.81 ± 0.59 *	4.60, 5.02	0.06

* Significant between-group difference (*p* < 0.05); CI = confidence interval; SE = standard error.

**Table 5 healthcare-11-01555-t005:** Slope of Median Frequency (Hz/periods) for the six muscles.

	Brace		No Brace		Between-Condition Cohen’s *d*
	Mean ± SE	95% CI	Mean ± SE	95% CI	
Slope of median frequency (%)					
Left cervical erector spinae	−0.051 ± 0.131	−0.320, 0.218	0.449 ± 0.167	0.108, 0.790	0.61
Right cervical erector spinae	0.068 ± 0.086	−0.168, 0.182	0.072 ± 0.057	−0.045, 0.189	0.16
Left upper trapezius	0.179 ± 0.096	−0.017, 0.375	0.116 ± 0.074	−0.036, 0.268	0.13
Right upper trapezius	−0153 ± 0.087	−0.331, 0.025	−0.088 ± 0.088	−0.269, 0.092	0.14
Left lower trapezius	0.379 ± 0.296	−0.226, 0.984	0.199 ± 0.102	−0.091, 0.408	0.15
Right lower trapezius	0.065 ± 0.071	−0.079, 0.209	0.036 ± 0.079	−0.125, 0.197	0.07

CI = confidence interval; SE = standard error; no between-group significant differences were found.

**Table 6 healthcare-11-01555-t006:** Mean Flexion Angles as measured by Inertial Measurement Units (IMUs) and a Camera Method.

	Brace						No-Brace						Between-Condition Cohen’s *d*
	1st minute		15th minute		30th minute		1st minute		15th minute		30th minute		
	Mean ± SE	95% CI	Mean ± SE	95% CI	Mean ± SE	95% CI	Mean ± SE	95% CI	Mean ± SE	95% CI	Mean ± SE	95% CI	
*IMU Measurements*													
Cervical spine	8.7 ± 1.6	5.5, 12.0	8.7 ± 1.6	5.4, 12.0	7.5 ± 1.6	4.2, 10.9	8.6 ± 1.4 *	5.8, 11.4	6.2 ± 1.6 *	3.0, 9.5	7.3 ± 1.4	4.4, 10.2	0.12
Thoracic spine	2.5 ± 1.2	−0.3, 5.0	4.5 ± 1.4	1.7, 7.4	4.7 ± 1.3	5.2, 7.3	4.2 ± 0.7	2.7, 5.7	7.0 ± 0.9	5.1, 8.9	6.5 ± 0.9	4.6, 8.3	0.02
Lumbar spine	2.4 ± 7.5	−6.3, 24.4	0.1 ± 1.5	−2.2, 3.9	6.5 ± 1.0	−1.6, 2.3	1.6 ± 1.0	-4.3, 3.7	2.5 ± 1.6	−7.2, 5.8	1.8 ± 1.7	−1.6, 5.2	0.05
*Digital Photograph Measurements*													
Forward head angle	51.0 ± 1.2	48.6, 53.5	52.7 ± 1.3	50.2, 55.3	52.8 ± 1.5	49.8, 55.9	50.6 ± 1.1	48.3, 52.9	51.3 ± 1.1	49.0, 53.5	52.6 ± 1.3	50.0, 55.3	0.03
Forward shoulder angle	45.5 ± 3.3	38.7, 52.4	46.0 ±3.6	38.8, 53.2	45.4 ± 3.6	38.1, 52.8	52.0 ± 2.7	46.6, 57.4	52.3 ± 2.7	47.2, 58.3	51.6 ± 2.8	45.9, 57.4	0.12

* Significant within-group difference between two timepoints (*p* < 0.05); CI = confidence interval; SE = standard error. However, the largest between-condition difference was only 2°.

## Data Availability

Relevant data supporting reported results can be made available from the corresponding author on request.

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
