# Peer review of "Effects of Using a Shoulder/Scapular Brace on the Posture and Muscle Activity of Healthy University Students during Prolonged Typing—A Randomized Controlled Cross-Over Trial"

_healthcare, 2023, doi:10.3390/healthcare11111555_

Round 1
Reviewer 1 Report
Manuscript shows the description of the brace using during typing task in university students assessed by SEMG. I have some considerations for you.
Introduction:
Topic is well introduced but few information about the braces are showed to justify your topic selection.
Methodology:
Participant section should start with the second paragraph and after that the sample size calculation.
Line 84: add a full stop before “The exclusion…”
To seat without backrest is not a “normal” situation. Why did you choose this position if it is a force posture? It could bias the results.
Line 122: add the first bracket to FSAs
Line 124: “could choose to sit or lie down” please explain the difference between this sit and the sit position during typing task. Is it the same?
How long was the break?
I read about the angle of the knees and other joints to perform the typing task, was the high of the screen or the chair adjusted to each participant?
Results
Line 241: “Fifteen males and 15 females!” please use the same format for numbers (number or letter).
Table 6 is really difficult to read.
Discussion
The strengths and limitations of the brace use must be exposed.
I am not sure if your data support or not the use of braces. In general braces, collars, wristbands… are contraindicated because the musculature lose their strength. For that reason, I do not completely understand the objective of the study. Please clarify the objectives and conclusions; besides, add some information about braces use in discussion section
Author Response
"Please see the attachment."

Reviewer 2 Report
This is a very well written paper on quantifying the immediate effects of scapular bracing on pain, fatigue, cervicothoracic posture, and the muscle activity of the neck and upper back.
1) The authors may want to add "Background" section by including more recent papers and discuss the importance of current study.
2) I would prefer a heading "limitations and future research." There are some limitations and directions that need to be addressed.
3) The authors may want to extend "Conclusion" section.
There are some grammatical errors throughout the manuscript, minor editing of English language is required.
Author Response
"Please see the attachment."

Reviewer 3 Report
Thank you for the opportunity to review this manuscript, below are my considerations:
Title:
- As in the whole study, I believe that the term asymptomatic does not fit the study well, since young people felt pain, that is, they were symptomatic. I believe the authors could change the term throughout the study to healthy college students, as they used a few sentences in the article;
- I suggest that the authors replace the current term: A randomized crossover study, by: A randomized controlled cross-over trial;
Abstract:
- Where is the purpose of the study?
- Insert the type (design) of the study, a randomized study is not a study design;
- When the authors report in the abstract that small improvements in shoulder posture were observed, this information is very vague and leaves room for different interpretations. What were those improvements? Better alignment? Better mobility? Better strength and function? Authors need to make it clear what these improvements were and in what aspects they occurred.
- I suggest inserting the p-value of all outcomes, including those in which there were no significant differences;
- What outcome did the authors want to assess with the electromyographic assessment? It was confusing, because they put the variables of this evaluation and not the outcome, I suggest inserting the name of the outcome;
- The authors named the outcome trapezius muscle activity and at various points in the study they mention this outcome as demand, muscle activity... I suggest that the authors standardize a single name for this outcome throughout the study.
- In the conclusions of the abstract, authors should only conclude what they found, if they wish to provide any suggestions for future studies, use the end of the discussion for this.
Introduction
- In the second paragraph of the introduction add the authors' names to their reference number, example: Choi and colleagues [14]....
- I believe that there is no connection between the second and third paragraphs, the authors end the second paragraph by saying that a good posture of the spine helps to better distribute loads in that location... and start the next paragraph by talking about an intervention. ... I believe that the authors can let find a better connection between the two paragraphs making the text fluid;
- In the third paragraph, in the first three lines, the authors mention a phrase, but there is no citation for the phrases, please insert;
- In the third paragraph of the introduction add the authors' names to their reference number, example: Lee and colleagues [15] ....
- What is this scapular posterior tilt exercise described in Lee's study? The scapula performs six movements: elevation/depression, retraction/protraction also called (adduction/abduction) and the up and down rotations. Therefore, I did not understand what exercise this would be, the authors need to clarify this.
- When the authors mention that only one investigation has studied the proposed theme and mention the reference [16], the way it is written, it is almost impossible to perceive that the information that comes immediately below refers to that study. I believe the authors need to make this paragraph more direct. After mentioning [16] like this, these authors found ... to make it clear to readers that the information below refers to the study you mentioned above, as the authors do not make any further citations in the paragraph.
Methods
- What is the study design? Authors need to insert the study design;
- What are the inclusion criteria? The authors mention the criteria for the exclusion of volunteers, but for their inclusion in the study what these criteria were, this needs to be included in the article;
- Were the students sedentary or physically active?
- As a clinical trial, the intervention protocol must have been published on an international platform, such as, for example, Clinical Trials, for an investigation of the protocol previously published on the platform and the final protocol contained in the article. The authors did not mention anything about this. I suggest that authors include the protocol registration number of this clinical trial for their analysis. In case the authors have not registered the protocol of this clinical trial, this must be mentioned as a serious limitation of this study;
- The methods are very confusing, the authors mention the steps and then go back to mention the steps already mentioned, that is, the description of the methods does not follow a logical and temporal sequence of how the steps of this study were carried out. I suggest that the authors modify/adjust the entire method of this article, demonstrating in the text a logical and temporal sequence of how all stages of this study were carried out, for example, it is not possible to describe the intervention and only after the intervention mention the evaluations. The evaluations took place in two moments, one of them before the intervention, so for those who read the article it is confusing. I ask the authors to be more careful in writing the study methods.
- If there was no blinding of the outcome evaluators, as it seems there was not, this is also a limitation of the study and should be included in the limitations of the study reported by the authors at the end of the discussion;
Results
- Where is the sample characterization table? When we are going to carry out a study comparing two or more groups, we need to know if these groups are comparable to each other, and this becomes possible through the sample characterization table, which contains the number of volunteers in each group and the data of their individual and physical characteristics. I suggest authors include the sample characterization table.
- Fix topic 3.2
- I couldn't understand table six, why are there some body regions without data?
Discussion
- In the discussion, the authors mention this finding: That said, the significantly lower LT activity during the brace trial might indicate that the brace provided passive stiffness to share the scapular stabilization role of LT so that the LT could relax temporarily. However, they do not bring physiological foundations to this discussion that justify how this could have occurred. I believe that authors should look for studies that justify their findings;
- The same occurs in the discussion of the cervical angle, the authors mention a study [16] that found similar results, but physiologically and/or biomechanically they do not bring any discussion about the finding. The sitting posture is undoubtedly one of the greatest villains for the spine, so adaptations and biomechanical errors are common in the spine during this position. Furthermore, the authors cannot forget the relationship between man and machine and man and furniture. Therefore, keyboard height, screen height, table height, the type of chair, with or without height adjustments and back support, are aspects that can influence the position of the spine. These aspects could justify this finding and the authors ignored it, I suggest a much more reasoned discussion on these aspects to the authors;
- The seventh paragraph of the discussion, in which the authors defend the use of orthoses, lacks much scientific support, since the discussion does not have well-founded physiological and/or biomechanical elements that guarantee the reader its use. I suggest that the authors improve the discussion by bringing evidence that provides a theoretical-scientific physiological and/or biomechanical model for the use of these orthoses, above all, due to the small size of the effect;
- The non-blinding of the outcome evaluators must be included as a limitation of the study, as well as the non-registration of the protocol, if it did not occur;
- Ergonomic implications: I suggest that authors include regular physical exercise with an emphasis on body posture as a therapeutic option for future trials. Even clinical trials that use control groups that did not perform any exercise are considered biased, so the ideal thing is for the clinical trial to compare two or more interventions and physical exercise is essential to prevent these musculoskeletal problems in these populations, so I suggest to authors who suggest regular physical exercise with an emphasis on body posture as a comparative therapeutic option for these devices to be analyzed in the future.
- Conclusions, the authors do not bring the conclusion of the study, they bring the objective, they provide some suggestions, but where is the conclusion of the study? Reinforcement once again, in the conclusions of the study the authors should only conclude what they found in that investigation, if they wish to provide any suggestions for future studies, use the end of the discussion for this.
Author Response
"Please see the attachment."

Round 2
Reviewer 1 Report
Dear Authors,
Thank you very much for taking into account my suggestions, I have a better understanding of your manucript.
Reviewer 3 Report
Congratulations to the authors, they did an excellent job, I believe that now the text of the article is clearer and more understandable for readers, in addition to the arguments of the discussion becoming more solid and scientifically robust. Congratulations once more.